# MIGHT: Statistical Methodology for Missing-Data Imputation in Food Composition Databases

**Gordana Ispirova** [1,2,*] **, Tome Eftimov** [1] **, Peter Korošec** [1] **and Barbara Koroušić Seljak** [1,3]

1   Computer Systems Department, Jožef Stefan Institute, Jamova cesta 39, 1000 Ljubljana, Slovenia;
    tome.eftimov@ijs.si (T.E.); peter.korosec@ijs.si (P.K.); barbara.korousic@ijs.si (B.K.S.)
2   Jožef Stefan International Postgraduate School, Jamova cesta 39, 1000 Ljubljana, Slovenia
3   School of Engineering and Management, University of Nova Gorica, Vipavska 13,
    5000 Nova Gorica, Slovenia
*   Correspondence: gordana.ispirova@ijs.si; Tel.: +386-14773519

**Abstract:** This paper addresses the problem of missing data in food composition databases (FCDBs). The missing data can be either for selected foods or for specific components only. Most often, the problem is solved by human experts subjectively borrowing data from other FCDBs, for data estimation or imputation. Such an approach is not only time-consuming but may also lead to wrong decisions as the value of certain components in certain foods may vary from database to database due to differences in analytical methods. To ease missing-data borrowing and increase the quality of missing-data selection, we propose a new computer-based methodology, named MIGHT - Missing Nutrient Value Imputation UsinG Null Hypothesis Testing, that enables optimal selection of missing data from different FCDBs. The evaluation on a subset of European FCDBs, available through EuroFIR and complied with the Food data structure and format standard BS EN 16104 published in 2012, proves that, in more than 80% of selected cases, MIGHT gives more accurate results than techniques currently applied for missing value imputation in FCDBs. MIGHT deals with missing data in FCDBs by introducing rules for missing data imputation based on the idea that proper statistical analysis can decrease the error of data borrowing.

**Keywords:** food composition databases; nutrient values; missing data; missing-data imputation; data borrowing; null hypothesis testing

## 1. Introduction

In food chemistry, chemical properties and interactions of food components are studied. Many chemical components, such as nutrients, occur in foods naturally and most of them contribute to balanced diet and eating pleasure. Components, called functional chemicals, play an important role in food production and preservation. Moreover, they can be effectively applied in the treatment and prevention of diseases. In foods, however, there are also chemicals with toxicological properties, some of which can cause harmful effects in humans and animals. Although it is very important to know which chemicals are built in foods, most countries still lack complete sets of food composition data (FCD). FCD is presented as a detailed set of information about the chemical components of foods, providing values for energy, nutrients and other bioactive components of foods (basic data elements), as well as food classifiers and descriptors (metadata). This type of data is available in Food Composition Databases (FCDBs) [1]. One of the most well-known applications of FCD is the nutrient-intake assessment at individual, regional, national or international level. FCDBs represent fundamental information resources for Food Science, however, they are also used in other public-health domains since the food industry, legislation and consumers all need and/or use FCD [2].

Two of the main limitations of FCDBs are: (i) variability in data between countries [3]; and (ii) incomplete coverage of foods or nutrients and bioactive components leading to missing data [4,5]. The variability in FCD between countries is a problem that occurs because different FCDBs may use different metadata to describe the same data. This problem can be solved by data harmonization, which is the process of bringing together data of varying file formats, naming conventions, and columns, and transforming them into one cohesive dataset, or in this case matching the same food items or nutrient descriptions from different FCDBs considering their matadata (i.e., classifiers and descriptors). Recently, normalization of short text segments (e.g., names or descriptions of nutrients) was proposed using two approaches: (i) standard text similarity measures; and (ii) a modified version of Part of Speech (POS) tagging probability weighted method [6–10].

The second limitation are missing data, which distort the integrity of the database. Missing FCD must never be assigned a zero value [5], which is a rule for FCDBs because a nutrient value can be zero [1]. A good quality FCDB should aim towards minimizing the amount of missing data. When there are no analytical values for missing FCD, the most used approach for resolving missing data is borrowing data from tables and databases from other countries, where a citation back to the original source may or may not be possible [11].

The FCDBs currently available in Europe and worldwide contain compositional values of differing quality, reflecting the various ways in which they were obtained [12–14]. They differ in basic data elements that are available, the metadata used to describe them, and in the quantity of the FCD that is held in the database. However, if FCD is intended to be used internationally, it must be of consistent and compatible quality, so that it can be used collaboratively between individuals and countries.

In FCDBs, data types and sources are identified by codes as done in many countries and by references [12–14]. These include, in order of preference [15]:

- Original analytical values: The data are taken from published literature or unpublished laboratory reports, whether or not prepared explicitly for the purpose of compiling the database.
- Estimated values: Estimates are derived from analytical values obtained for a similar food or for another form of the same food.
- Calculated values: Data are derived from recipes, calculated from the nutrient contents of the ingredients and corrected for preparation factors [16].
- Borrowed values: The data are taken from other tables and databases.

Since the food supply has evolved, and the demand for nutritional and bio-active components grows, relying only on chemical analysis when compiling FCDBs has become almost impossible. According to current "rules", or rather suggestions, missing values should be borrowed from a FCDB, which contains the foods and nutrients of interest and/or is compiled from geographically similar countries [17]. In practice, the borrowing of data for imputation of missing values is performed either using a FCDB well known for its data quality or by using several FCDBs and using either an average or median value of their values [18]. In our case, the average is the arithmetic mean, which is the sum of a collection of data values divided by the number of data values in the collection. The median is the value separating the higher half from the lower half of a data sample. For example, for a dataset, it may be thought of as the "middle" value [19]. Using these approaches, however, can lead to inaccuracies. Foods also exhibit variations in composition, and the composition of any given single food sample cannot be accurately predicted. The prediction accuracy is also constrained by how data are maintained in a FCDB (as averages or best estimates, for example). Clearly, to increase the quality of FCDBs, there is a need for methods that can be used either for imputation or calculation of missing FCD.

In this paper, we present a semi-automatic computer-supported methodology for borrowing missing nutrient values by generating rules using null hypothesis testing. The end result for a missing value of a specific nutrient in a food from a given country is an average or median value from the values of the set of countries whose FCDBs are eligible for borrowing. Several experiments were conducted and the experimental results demonstrate that our methodology provides high accuracy

when compared with the most commonly used approaches. The rest of the paper is structured as follows. In Section 2, we give explanation of the data for the experiments and the background literature for the methodology. The experimental results are presented in Section 3. A comparison with state-of-the art methodologies, time efficiency as well as further discussions about the methodology are given in Section 4. Finally, in Section 5, a summarization of the importance of this methodology and directions for future work are presented.

## 2. Materials and Methods

This section contains the data used in our experiments, how it is obtained, and the format in which it is used. After this subsection, the background that is needed to understand the methodology in this paper is presented. Here, we give an overview of the current techniques for borrowing FCD, and the current state-of-the-art approaches for obtaining missing values. The final subsection is a detailed explanation of the methodology itself.

### 2.1. Food Composition Data

There are several international bodies, organizations and projects working on food composition. One of them is EuroFIR (European Food Information Resource Network) AISBL, which is an international, non-profit association under the Belgian law [20]. Its purpose is to develop, publish and exploit food composition information and promote international cooperation and harmonization of standards to improve data quality, storage and access. One of the aims of the EuroFIR Network of Excellence [20] was to develop the standard "Food data structure and format standard" BS EN 16104:2012 [21], which can be used as a framework for compiling and disseminating FCD that is comparable and unambiguous with respect to the identity and description of foods, components and compositional values [20,22]. Each EuroFIR member country has compiled one or more national FCDBs that are published online and are accessible by the public through the FoodEXplorer tool [23].

For the purpose of this study, we chose to collect data from the national FCDBs of 10 countries: Italy (IT), United Kingdom (UK), Switzerland (CH), Sweden (SE), Slovenia (SI), Belgium (BE), Denmark (DK), Netherlands (NL), United States of America (USA), and Canada (CA). In the process of collecting, we also considered the following key points:

1.  Choosing method type: As mentioned, there are several types of FCD with differing quality. Terms for documenting the method used to obtain a compositional value, including analysis, calculation and imputation, are presented in the EuroFIR Method Type Thesaurus [24]. In the thesaurus, the terms are organized in a hierarchy and for each method a short abbreviation with a full description used for classification is given. Since we are proposing a method for borrowing data for imputation of missing FCD between countries, it is also important that the data are relevant and genuine, therefore, we only included data whose method type is listed as:

    *   "A" (Analytical result/s): The value is based on an analytical result or a statistic of multiple measurements of the same food sample (replicates).
    *   "AG" (Analytical, generic): The value is known to be analytical, but no further information on the nature of analysis is available, whether the value derives from the same or different statistical distributions.

2.  Choosing food groups: For this study, we decided to include the food groups: "Fruits", "Vegetables" and "Meats".
3.  Choosing foods that belong to the selected food groups: In this stage of the study, we decided to include only simple, raw foods for the convenience of easily selecting the same foods from different countries, and because, generally, there more available data worldwide for simple, raw foods.

4.  Choosing nutrients: This step is highly dependent on the available data. Given the selections from the previous steps, we chose nutrients with the most data available. For the evaluation of the proposed methodology, for the food group "Fruits", the nutrients chosen are Sodium (Na) and Potassium (K); for the food group "Vegetables", Sodium (Na); and for the food group "Meats', Protein (PROT).

5.  Choosing countries: The last step is to manually choose specific countries for each of the selected food groups. For this purpose, we constructed a binary matrix for each nutrient in each food group. The matrix rows correspond to the selected foods belonging to one food group, the columns are the identification numbers of the national FCDBs in EuroFIR FCDB, "1" represents the presence and "0" the absence of the required nutrient value. Using this matrix, we can find the countries, i.e., the national FCDBs for which the methodology can be evaluated. After the data collection process, before assembling it all in one table, the last step is to ensure that all the values are in the same measure unit; if not, the needed unit conversions are made. Table 1 presents a binary matrix for the nutrient Potassium (K) in the food group "Fruits". The end result is a data table that contains values of the selected nutrient in the foods from the corresponding food group. In Table 2, a general representation of the data used in our experiments is given, where *m* is the number of selected countries and *n* the number of considered foods per food group.

**Table 1.** Binary matrix for selecting which national FCDBs to include.

| Food | NO | FR | IT | UK | CH | SE | ES | BE | DK | USA | CA | SI | NL | CZ | AU | PT |
|---|---|---|---|---|---|---|---|---|---|---|---|---|---|---|---|---|
| Apple | 1 | 1 | 1 | 1 | 1 | 1 | 0 | 1 | 1 | 1 | 1 | 1 | 1 | 0 | 1 | 0 |
| Banana | 0 | 1 | 1 | 1 | 1 | 1 | 0 | 1 | 1 | 1 | 0 | 0 | 1 | 0 | 1 | 0 |
| Blueberry | 0 | 1 | 1 | 1 | 1 | 1 | 0 | 1 | 1 | 1 | 1 | 1 | 1 | 0 | 1 | 1 |
| Cherry | 0 | 1 | 1 | 1 | 1 | 0 | 0 | 1 | 1 | 1 | 1 | 1 | 1 | 0 | 0 | 0 |
| ⋮ | ⋮ | ⋮ | ⋮ | ⋮ | ⋮ | ⋮ | ⋮ | ⋮ | ⋮ | ⋮ | ⋮ | ⋮ | ⋮ | ⋮ | ⋮ | ⋮ |

**Table 2.** Generalized for values from one nutrient in several foods from several FCDBs.

| Food/FCDB | $Country_1$ | $Country_2$ | $Country_3$ | ... | $Country_m$ |
|---|---|---|---|---|---|
| $Food_1$ | $Value_{11}$ | $Value_{12}$ | $Value_{13}$ | ... | $Value_1 m$ |
| $Food_2$ | $Value_{21}$ | $Value_{22}$ | $Value_{23}$ | ... | $Value_2 m$ |
| $Food_3$ | $Value_{31}$ | $Value_{32}$ | $Value_{33}$ | ... | $Value_3 m$ |
| ⋮ | ⋮ | ⋮ | ⋮ | ⋮ | ⋮ |
| $Food_n$ | $Value_{n1}$ | $Value_{n2}$ | $Value_{n3}$ | ... | $Value_{nm}$ |

## 2.2. Related Work

This subsection starts with the traditional approaches for borrowing data for imputation of missing FCD in the context of FCDBs, and continues with a chosen state-of-the-art methodology for missing value imputation in general.

### 2.2.1. Traditional Approaches for Borrowing FCD

Although food composition data may appear to be simply a collection of foods with the nutrients listed alongside them, to ensure that the values reported are used correctly, there are actually a number of considerations that users should take into account:

*   variability of the composition of a food item;
*   misunderstanding of nutrient definitions;
*   use of an incorrect conversion factor; and
*   use of a nutrient interchangeably with its sub-types.

When using FCD on its own, the limitations that should be considered include the inherent variability in the composition of a food item, the number of food items and range of nutrients covered

by a database. Errors that may be associated with using a FCDB include: an understanding of nutrient definitions such as available carbohydrate (excluding fiber) versus total carbohydrates (including fiber) and calculation errors for fatty acid values. This commonly includes the use of an incorrect conversion factor, e.g., the use of fatty acids per 100 g of total fatty acids instead of per 100 g of the food item, or calculation of vitamin A intake without considering the pro-vitamin A compounds (alphacarotene, beta-carotene, and betacryptoxanthin). Similarly, interchangeable use of vitamin E and alpha-tocopherol is another common error (vitamin E actually includes alpha-, beta-, delta- and gamma-tocopherols and tocotrienols).

Nowadays, many countries have their own national FCDBs that are readily available online [20]. When using the food composition data of another country, there are few things that we must be observant of:

- differing cultivators, soils, climates and agricultural practices;
- differing food production and processing practices;
- variation in the composition of recipe ingredients with the same name; and
- variation in food products available.

Each of these factors can have significant impact on the micro-nutrient composition of food items. FCD may differ between countries as a result of different things such as varied definitions of nutrients, analytical methods (e.g., fiber), nutrient calculations (e.g., energy), the reference quantity of foods contained in the database (per 100 g vs. per serving) and the nutrient values that are available for the foods. The treatment of missing values may vary between countries despite efforts to adjust the practices. When selecting another FCDB to borrow values from, the following must be taken into account [17,18]:

1. The FCDB should contain up-to-date FCD.
2. The FCDB should be of high quality (foods and components are well defined and described, using appropriate analytical methods and nutrient definitions, a variety of foods types are provided, e.g., raw, cooked, recipes, brand name, fortified, and supplement).
3. The FCDB should contain foods and nutrients of interest.
4. The FCDB should come from a country that is similar in terms of geographic location, agriculture, food production, recipes and food processing.

Often, these rules are not taken into account, simply because the FCDB that fulfills the first, second and fourth rule does not fulfill the third rule, i.e., it does not contain the nutrients of interest, or simply one cannot find a database that fulfills the first three and it is in similar geographic location (the fourth rule). Because these "rules" are demanding and often without a possible solution, the users choose other ways of finding an eligible FCDB for borrowing. The most common solution is borrowing from the FCDB of one of the neighboring countries, or just relaying on a FCDB with a big span of data.

As a solution, users also choose to calculate average of values from multiple countries, and as of today this is considered to be the most appropriate and accurate method for borrowing data for imputation of missing values.

2.2.2. Non-Negative Matrix Factorization

Recently, non-negative matrix factorization (NMF) has been proven to be useful for missing value imputation in many applications in the environment, pattern recognition, multimedia, text mining, and DNA gene expressions. The roots of NMF can be traced back to the 1970s [25] and was initiated and studied extensively by Paatero and Tapper [26].

NMF [27,28] is a group of algorithms in multivariate analysis and linear algebra where a matrix $V$ is factorized into two matrices $W$ and $H$, where all three matrices have no negative elements. This property of non-negativity makes the resulting matrices easy to inspect.

Non-negative matrix factorization (NMF) has previously been shown to be a useful decomposition for multivariate data. The authors of [29] interpreted the factorization in a new way and used it to generate missing attributes from test data. They provided a joint optimization scheme for the missing attributes as well as the NMF factors. In addition, they proved the monotonic convergence of their algorithms, and presented classification results for cases with missing attributes.

### 2.3. MIGHT—Missing Nutrient Value Imputation Using Null Hypothesis Testing

Working with FCDBs has its downsides. As analytical FCD is very valuable, relevant, difficult and expensive to obtain, and missing values are a very common problem in general in FCBDs, it is evident that in our case we will be working with small datasets. As we are working with food groups, i.e., selecting foods that belong to a specific food group, an important part here is to note that each set of values formed for one food group must consist of a minimum number of foods $n$. This implies finding existing, non-null values for one nutrient in $n$ foods that belong to a certain food group where all values are type "A" (Analytical result/s) or "AG" (Analytical, generic). To find the minimum number $n$, we must go further with our methodology.

The idea of hypothesis testing used with biological and chemical data is presented in [30,31]. Modeling based on null hypothesis testing has been previously used in a lot of application in the area of economics and information systems [32]. Alongside effect size, confidence intervals, and $p$-values are part from frequentist statistics, albeit with increasing movements towards Bayesian approaches, for much of the health sciences. MIGHT follows the idea of frequentist statistics and involves statistically comparing values from different FCDBs, and based on the statistical significance between them creating rules for borrowing. One common approach for making a statistical comparison is null hypothesis testing. To perform null hypothesis testing, the number of instances, $n$, in the datasets should satisfy a requirement for a minimum sample size, $r$, for an appropriate statistical test. This implies that $n \geq r$.

Since we are working with small datasets, to build more robust models, we decided on leaving one instance out, then building models for the remaining instances, and at the end aggregating all the models to obtain the final rules for imputation. To achieve this, we require an additional instance: $n \geq r + 1$. Lastly, because this methodology needs to be tested on new, unseen instances, we require one more instance in our final set: $n \geq r + 2$. To generate rules for imputation between countries, the number of countries, $m$, involved in the comparison should be $m \geq 3$. For example, if we have only two countries, where one should borrow from the other, the user does not have any option for imputation, so we need at least three. This also comes from the fact that, using the traditional approaches, in most cases, users calculate either average or median from several countries.

The starting point is a dataset such as the one presented in Table 2, with the restrictions of $n \geq r + 2$ and $m \geq 3$. Before an appropriate statistical test is applied, some assumptions (i.e., conditions) must be checked. The most common assumptions checked are independence, normality of the data, and homoscedasticity of the variances [33]. In probability theory, two events are independent (i.e., statistically or stochastically independent) if the occurrence of one does not affect the probability of occurrence of the other. Similarly, two random variables are independent if the realization of one does not affect the probability distribution of the other. A random variable with Gaussian distribution is said to be normally distributed with some average value, $\mu$, and standard deviation, $\sigma$. Normality tests assess the likelihood that the given dataset comes from a normal distribution, which can be checked by using one of the following tests: Kolmogorov–Smirnov [34], Anderson–Darling [35], Shapiro–Wilk [36], and D'Agostino–Pearson [37] test. The assumption of homogeneity of variance is that the variance within each of the datasets is equal. To test for homogeneity of variance, several statistical tests exist, such as Cochran's test [38], Levene's test [39], and Barlett's test [40]. The most common assessment for homogeneity of variance is the Levene's test, which uses an F-test to test the null hypothesis that the variance is equal across groups (i.e., datasets). With regard to obtained results for checking of assumptions, an appropriate statistical test must be selected. Each statistical test has its own required assumptions. Parametric tests have been commonly used in analyses, however, as a general

rule, a non-parametric test is less restrictive than a parametric one, although it is less robust than a parametric when data are well conditioned. For example, if a violation of homogeneity of variance occurs, it is likely that conducting the non-parametric equivalent of the analysis is more appropriate.

After selecting the appropriate statistical test, we then test the null hypothesis $H_0$. In our case:

- $H_0$: There is no effect/difference between the separate datasets.
- $H_1$: There is an effect/difference between the separate datasets.

Each statistical test has a test statistic, which is a mathematical formula used to obtain a value using the collected data samples. One way is that this value is then compared with a value from a table that contains information about the distribution of the test statistic. These tables contain extreme values (critical values) of the test statistic that are highly unlikely to occur if the null hypothesis is true (depending of the type I error rate of the test). Before obtaining a value from such a table, we first need to specify a level of significance ($\alpha$). This value is a probability threshold below which the null hypothesis will be rejected. In [41], the authors detailed the selection of the level of significance ($\alpha$), and the relation between $\alpha$ and the sample size. From the work in [42], we can get more details on the choice of $\alpha$ and its connection with sample size. Instead of using critical values, nowadays, in frequentist statistics, a $p$-value (i.e., probability value or asymptotic significance) can be used, which is the probability for a given statistical model that, when the null hypothesis is true, the statistical summary would be greater or equal to the actual observed results. A smaller $p$-value indicates that the null hypothesis is rejected, while a larger $p$-value indicates that the null hypothesis is not rejected.

After applying the appropriate statistical test, we obtain the result of testing the null hypothesis, i.e., the $p$-value, which is then compared with the chosen significance level $\alpha$ to see if the null hypothesis $H_0$ is rejected ($p < \alpha$) or not rejected ($p \geq \alpha$). If $H_0$ is rejected that means that the result is statistically significant, thus it is important from which country (or countries) the value is borrowed. Once the test rejects the null hypothesis, the detection of the specific differences among the countries can be made with the application of post-hoc statistical procedures, which are methods used for specifically comparing a country that is a subject of imputation with two or more countries. The post-hoc tests differ in the way they adjust the value in order to compensate for multiple comparisons [43]. The results obtained from the post-hoc test are again compared with the significance level, $\alpha$, and based on that a matrix of ones and zeros is generated, which represents the rules for borrowing:

- "0": $H_0$ is not rejected, indicating that the datasets can borrow interchangeably.
- "1": $H_0$ is rejected, meaning that the datasets cannot borrow from each other.

Figure 1 presents a flowchart of the methodology.

If we have $n$ instances (problems), we repeat the following steps $n$ times:

1. Leave one instance for testing.
2. On the $n - 1$ instances repeat $n - 1$ times:

    - Leave one instance out.
    - Check conditions for the $n - 2$ instances left.
    - Choose the appropriate omnibus test.
    - Check $p$-value:

        – If $p \geq \alpha$ — the null hypothesis $H_0$ is not rejected, indicating that each country can borrow from any other;
        – If $p < \alpha$ — the null hypothesis $H_0$ is rejected, so we need to continue with the appropriate post-hoc procedure for the previously selected omnibus test.

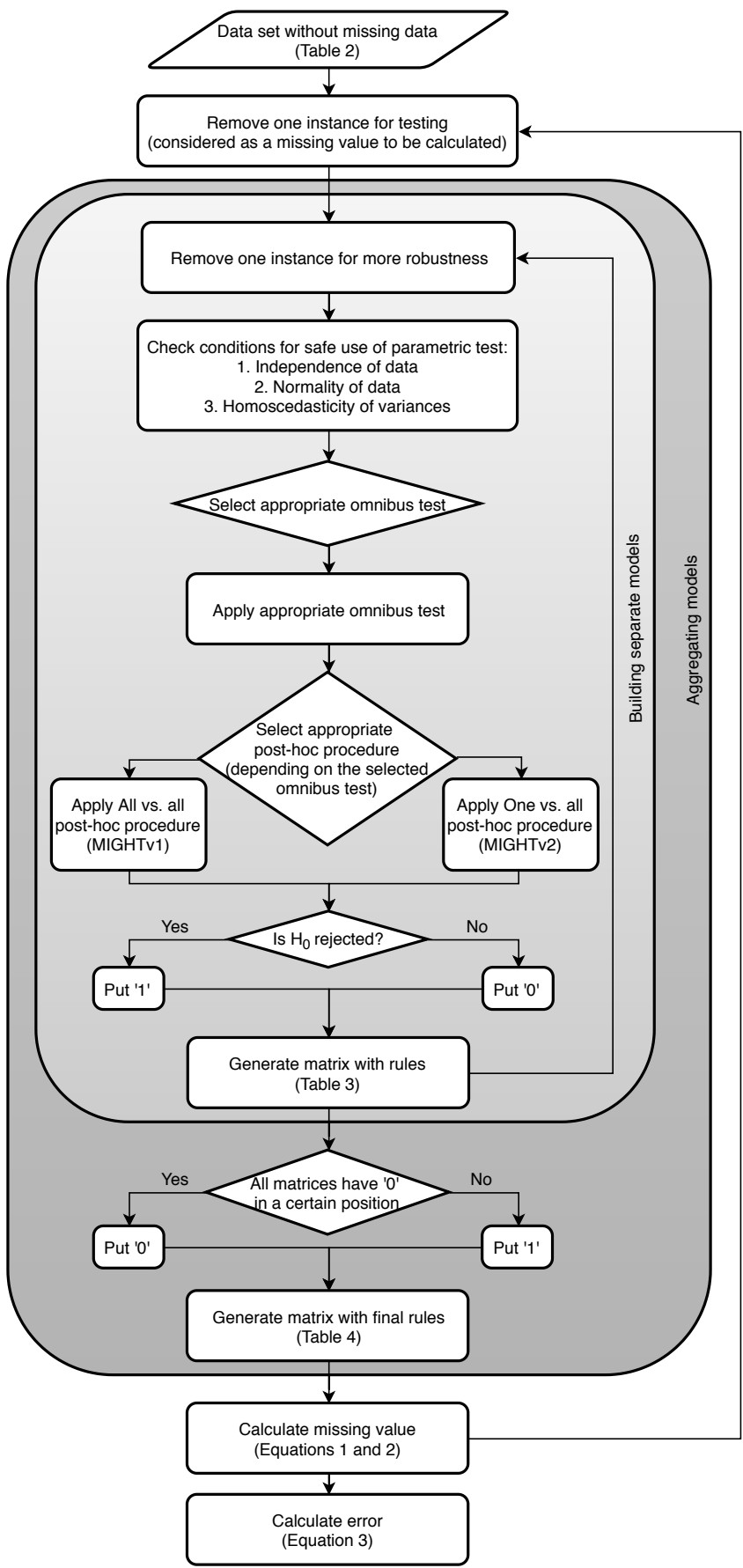

**Figure 1.** Flowchart of the methodology.

- Apply appropriate post-hoc test [44], and depending on the type of post-hoc procedure applied, the methodology is split into two versions:

  – MIGHTv1: Apply All vs. all type of post-hoc test (simultaneous) on the set and generate a matrix with the result values. All vs. all type of post-hoc test is a multiple comparison type of test, multiplicity or multiple testing problem, which occurs when one considers a set of statistical inferences simultaneously or infers a subset of parameters selected based on the observed values [45,46].

  – MIGHTv2: Apply One vs. all type of post-hoc test (step-down) $n - 2$ times and generate a matrix with the result values. One vs. all type of post-hoc test is a multiple test procedure of the sequentially rejective type, i.e., hypotheses are rejected one at a time until no further rejections can be done [47–49].

- Compare every value from the two generated matrices from the post-hoc testing with the selected significance level $\alpha$ and generate new matrices with rules (separate matrix from the MIGHTv1 and separate matrix from the MIGHTv2 results). The matrices with rules are filled in according to:

  – If $H_0$ *is not rejected*, then put "0".This indicates that from the specific model we have gained information that country $x$ can borrow from country $y$.
  – If $H_0$ *is rejected*, then put "1". This means that from the specific model we have gained information that country $x$ cannot borrow from country $y$.

  Table 3 gives a representation of these matrices, from which we can see that $Country_1$ cannot borrow from $Country_2$, and $Country_2$ cannot borrow from $Country_m$.

**Table 3.** Example table of matrix with rules obtained from one model, where $m$ is the number of countries, 0 indicates that according to this model the countries from the specified row and column can borrow from each other, while 1 indicates the opposite, i.e., they cannot borrow from each other, and / indicates "Not applicable".

| Country | $Country_1$ | $Country_2$ | ... | $Country_m$ |
|---------|-------------|-------------|-----|-------------|
| $Country_1$ | / | 1 | ... | 0 |
| $Country_2$ | 1 | / | ... | 1 |
| ⋮ | ⋮ | ⋮ | ⋮ | ⋮ |
| $Country_m$ | 0 | 0 | ... | / |

3. Generate two final matrices with rules (one for MIGHTv1 and MIGHTv2) from all the separate matrices with rules. These matrices are generated according to:

- If all the *n-2* matrices have "0" in a certain position, then in the final matrix we put "0".
- If one or more of the *n-2* matrices has "1" in a certain position, then in the final matrix we put "1".

An example table of a sum from the rules from all the models is presented in Table 4. Here, "0" indicates that according to all models the countries from the specified row and column can borrow from each other, while everything else indicates the opposite, i.e., they cannot borrow from each other.

**Table 4.** Matrix formed from summing the rules from all the models, where $m$ is the number of countries, $n$ is the number of models, $a, b \in \mathbb{N}$, and $0 \leq a, b \leq n$. For MIGHTv1 $a = b$, for MIGHTv2 $a \neq b$.

| Country | Country$_1$ | Country$_2$ | ... | Country$_m$ |
|---------|-------------|-------------|-----|-------------|
| Country$_1$ | / | $n - a$ | ... | $n - a$ |
| Country$_2$ | $n - b$ | / | ... | $n - a$ |
| ⋮ | ⋮ | ⋮ | ⋮ | ⋮ |
| Country$_m$ | $n - b$ | $n - b$ | ... | / |

4.  Test the rules on the separated, unseen instance. The final rules are incorporated in metrics for calculating the missing value of the selected nutrient in any food from the corresponding food group for any of the given countries (Equations (1) and (2)).

$$Calculated_{average} = Average\left\{ values\ from\ FCDBs\ eligible\ for\ borrowing \right\} \tag{1}$$

$$Calculated_{median} = Median\left\{ values\ from\ FCDBs\ eligible\ for\ borrowing \right\} \tag{2}$$

5.  Compare the results from the rules with the actual value (Equation (3)).

$$Error = \mid Actual_{value} - Calculated_{value} \mid \tag{3}$$

## 3. Results

For the evaluation of MIGHT, we used the "scmamp" (Statistical Comparison of Multiple Algorithms in Multiple Problems) [50] package for the R programming language. The idea is that, when given a matrix with results of different algorithms for different problems, the package uses statistical tests and corrections to assess the differences between algorithms, which is the main idea behind MIGHT, except that we are not dealing with algorithm performances.

We evaluated our methodology using eight different experiments on four different datasets. For each dataset, to create the rules for borrowing, the methodology uses both MIGHTv1 and MIGHTv2. After creating the rules for each country, its missing value can be either an average or a median of the values of the countries that are eligible for borrowing. We also calculate the missing value as an average or a median from the value from all of the countries that are involved in the borrowing process. This means not only the countries that are eligible and are results from our methodology. We have done this for each country on each test instance from the dataset, and, in both scenarios, we calculate the absolute error between the borrowed value and the true analytical value.

For a better understanding of how the methodology works, we present an example dataset and present the evaluation process in detail. The selected dataset contains nutrient values collected for Potassium (K) in foods belonging to the food group "Fruits". The data collection process is as follows:

1.  Method type is chosen as "A" (Analytical result/s) or "AG" (Analytical, generic).
2.  Food group is "Fruits".
3.  In this example, the food group includes only raw foods, so the separate foods, i.e., fruits, are chosen aiming towards as much data as possible.
4.  The nutrient is Potassium (K), since it is common nutrient in fruit [51]. At the first stage, we collected data for 18 foods: apple, banana, blueberry, cantaloupe, cherry, grape, grapefruit, lemon, mango, melon, orange, peach, pear, pineapple, plum, raspberry, strawberry, and tomato.
5.  To choose the countries, a binary matrix is constructed. The rows are all the fruits that we have data for, and the columns are the identification numbers of the national FCDBs. Although this step is meant for choosing countries, we are also recursively choosing foods, because there is no point in keeping foods that have missing data for many countries.

After collecting the Potassium (K) data for the group "Fruits", we check that all the data are in the same measure unit (grams, milligrams, etc.), and if needed conversions are made. In this example, the dataset consists of data for 16 fruits extracted from 9 national FCDBs, i.e., $n = 16$ and $m = 9$. The next step is to apply our methodology to the data. First, we check the restrictions for the number of problems and datasets ($n \geq r$ and $m \geq 3$). Since $m = 9$, the condition for the number of datasets $m \geq 3$ is satisfied. The number $r$ depends on the statistical test used, so in order to find $r$ we first need to know what kind of statistical test to apply. Before testing the null hypothesis, the three conditions for choosing the appropriate test must be checked:

1. Independence: In our case, events are the processes used to obtain/measure the values of Potassium (K) in a fruit. Since we are working with data from different national databases, and we made sure to only include values measured analytically, the datasets in our example are independent.

2. Normality is checked using the Shapiro–Wilk test from the "stats" package [36]. The test is applied to each column of experimental data, i.e., for each dataset separately, resulting in not meeting the requirement for normality for all columns. After this, there is already one condition that is not satisfied, which means a non-parametric test must be used.

3. Checking for homoscedasticity at this point does not bring any contribution to the result, but since this is not the only experiment conducted for evaluation it must be mentioned. The homoscedasticity is checked by applying the Levene's test from the "lawstat" package [52]. The result from the test indicates that the condition for homoscedasticity is not satisfied.

Since our data do not satisfy the required conditions for the safe use of a parametric test, it is better to use an analogous non-parametric test. Because the set has more than two columns, we apply the Friedman test [53]. A requirement for the Friedman test is that the number of problems (in our case foods) must be greater than 10, so, for our methodology, $r > 10$ [43] and $n = 16$, which means that $n \geq r$. Further, we have 16 instances, which means the following is repeated 16 times:

1. One instance is removed for testing.
2. For each of the remaining 15 instances, repeat the following steps 15 times (i.e., perform leave one out validation):

   (a) One instance is removed.
   (b) Check conditions for the remaining 14 instances.
   (c) Apply Friedman test and obtain $p$-value.
   (d) The $p$-value is compared with the significance level, which in our case is chosen to be $\alpha = 0.05$ [54]. In all of the runs, we obtained a smaller $p$-value, $p < \alpha$. As mentioned above, this indicates that the null hypothesis is rejected, meaning that there is a statistical significance between the datasets, i.e., between the FCD from the different countries. Further, to see where this differences come from, we use two different post-hoc procedures, which makes our methodology branch out into two different versions:

   - MIGHTv1: when All vs. all type of post-hoc procedure is applied.
   - MIGHTv2: when One vs. all type of post-hoc procedure is applied.

   (e) Friedman test post-hoc procedure:

   - Nemenyi test [55] is applied as an All vs. all post-hoc procedure for the previously used Friedman test. The Nemenyi test provides a matrix with $p$-values for each pairwise comparison.
   - Holm procedure [47] of $p$-value adjustment is applied nine times, where each time a different country is selected as a control (One vs. all). The end result is a matrix

with *p*-values, where each row corresponds to the results obtained from the Holm procedure (i.e., the country that represents the row is selected as control country).

(f) Using the *p*-values obtained in these matrices, for each pairwise comparison we can check if the null hypothesis is rejected (1), or not rejected (0). If the null hypothesis is rejected, it means that there is a statistical significance between the FCD, while, if it is not rejected, it means that there is no statistical significance between the FCD. Table 5 gives an example of a matrix with rules for one model generated from MIGHTv1. In these tables, we can see that, for example, for a missing Potassium (K) value in a food from the food group "Fruits" for the Italian FCDB following the rules of MIGHTv1 we cannot borrow from the Canadian FCDB and following the rules of MIGHTv2 we cannot borrow from the FCDBs from Sweden, USA and Canada.

**Table 5.** Matrix with rules generated from one model form MIGHTv1 for Potassium (K) in foods from the food group "Fruits".

| Country | IT | UK | CH | SE | BE | DK | USA | CA | NL |
|---------|----|----|----|----|----|----|-----|----|----|
| IT  | 0 | 0 | 0 | 0 | 0 | 0 | 0 | 1 | 0 |
| UK  | 0 | 0 | 0 | 0 | 0 | 0 | 0 | 0 | 0 |
| CH  | 0 | 0 | 0 | 0 | 0 | 0 | 0 | 0 | 0 |
| SE  | 0 | 0 | 0 | 0 | 0 | 0 | 0 | 0 | 0 |
| BE  | 0 | 0 | 0 | 0 | 0 | 0 | 0 | 0 | 0 |
| DK  | 0 | 0 | 0 | 0 | 0 | 0 | 0 | 0 | 0 |
| USA | 0 | 0 | 0 | 0 | 0 | 0 | 0 | 0 | 0 |
| CA  | 1 | 0 | 0 | 0 | 0 | 0 | 0 | 0 | 0 |
| NL  | 0 | 0 | 0 | 0 | 0 | 0 | 0 | 0 | 0 |

(g) The two final matrices with rules are generated by sum the matrices obtained from each comparison using both versions, respectively. For this example, this type of matrix is presented in Table 6. In these tables, we can see that, for example, the rule that for a missing nutrient value of Potassium (K) in a food from the food group "Fruits" from the Italian FCDB following the rules from MIGHTv1 we cannot borrow from the Canadian FCDB has appeared in 12 comparisons.

**Table 6.** Matrix with final rules from MIGHTv1 for Potassium (K) values in foods from the food group "Fruits".

| Country | IT | UK | CH | SE | BE | DK | USA | CA | NL |
|---------|----|----|----|----|----|----|-----|----|----|
| IT  | 0  | 0 | 0 | 0 | 0 | 0 | 0 | 12 | 0 |
| UK  | 0  | 0 | 0 | 0 | 0 | 0 | 0 | 0  | 0 |
| CH  | 0  | 0 | 0 | 0 | 0 | 0 | 0 | 0  | 0 |
| SE  | 0  | 0 | 0 | 0 | 0 | 0 | 0 | 0  | 0 |
| BE  | 0  | 0 | 0 | 0 | 0 | 0 | 0 | 0  | 0 |
| DK  | 0  | 0 | 0 | 0 | 0 | 0 | 0 | 0  | 0 |
| USA | 10 | 0 | 0 | 0 | 0 | 0 | 0 | 0  | 0 |
| CA  | 12 | 0 | 0 | 0 | 0 | 0 | 0 | 0  | 0 |
| NL  | 0  | 0 | 0 | 0 | 0 | 0 | 0 | 0  | 0 |

3. The final rules from both scenarios are tested on the previously removed instance. As mentioned, the rules are incorporated in metrics for calculating the missing nutrient value in the selected food for any of the given countries. The chosen metrics are the median and average, which are calculated from the values of the FCDBs eligible for borrowing ('0' in all models). These new values are then compared with the true values (analytical values) of the test instances by calculating the absolute error (Equation (3)).

Until now, by following the "rules" for borrowing it is not possible to make the right decision from where to borrow, this is because there is no correct decision, and the most desirable scenario is to make the best decision for a given situation. The main benefit of using our methodology for missing value imputation is that the methodology provides countries that are eligible for calculating the missing value of interest. Using the nutrient values from those countries, descriptive statistics can be applied for missing value imputation.

To evaluate MIGHT, we compared our results with the averages and medians obtained without following the rules. In the food group "Fruits", we predicted the Potassium (K) value for each food (when it is used for testing) regarding the generated rules, assuming that we have the values from other countries. The initial step involved calculating the averages and medians of the nutrient values from all the countries in the study. Next, we calculated the error, i.e., their deviation from the actual nutrient value, and compared this with the error values obtained using the borrowing rules. From the nine countries tested for 16 foods, we made 144 predictions.

MIGHTv1 gives better results in 84.0% compared to calculating the nutrient value as an average from all nine countries (i.e., average with no rules). When we use median instead of average, MIGHTv1 also gives better results (84.7%) versus calculating the nutrient value as a median from all nine countries, without considering the rules for borrowing. MIGHTv2 provides better results in 66.94% of the cases compared to calculating the nutrient value as an average without considering the rules, and in 76.0% of the cases compared to calculating the nutrient value as a median wihout considering the rules. In all cases better results are measured as smaller absolute error.

The distributions of the absolute error for each version of our methodology and the common approach using average or median from all of the countries involved in borrowing are presented in Figures 2 and 3. In the figures, we can see that the distributions of the absolute errors for averages calculated with MIGHT and regular averages are different. The same is also true for the distributions of the absolute errors for medians calculated with MIGHT and regular medians. Comparing the distributions between averages calculated with MIGHT and regular averages (or medians calculated with MIGHT and regular medians), it becomes obvious that MIGHT provides smaller absolute errors. In addition, the interquartile range, which is a measure of variability (statistical dispersion), is smaller, indicating a small variability in the absolute error. This is not the case when using regular averages or medians.

We selected absolute error for comparison in order to provide a general estimation for the magnitude of the error, without considering if the value is above or below the true analytical value. This is done because nutrients are substances used by an organism to survive, grow, and reproduce. All organisms, including humans, obtain nutrients from the surrounding environment and require some of the nutrients (i.e., macronutrients, such as fats, carbohydrates, proteins and water) in relatively large amounts (grams), while other nutrients (i.e., micronutrients, such as vitamins and minerals) are needed in smaller amounts (milligrams, micrograms or even nanograms).

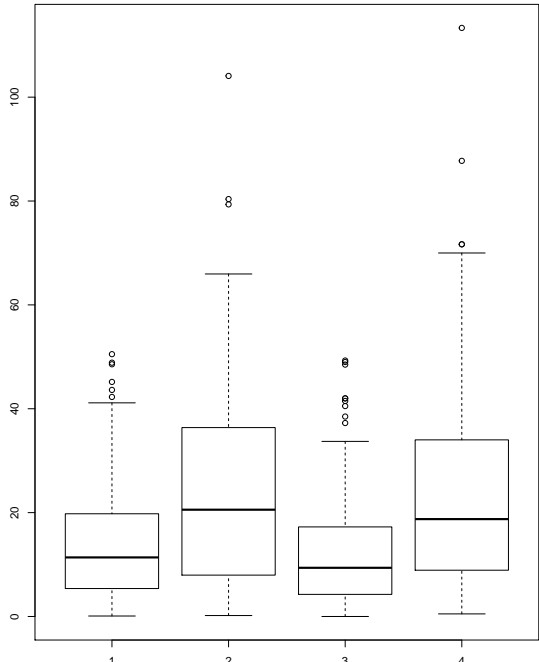

**Figure 2.** Distribution of absolute error of: MIGHTv1 average, regular average, MIGHTv1 median and regular median calculated for the Potassium (K) content in foods from the food group "Fruits".

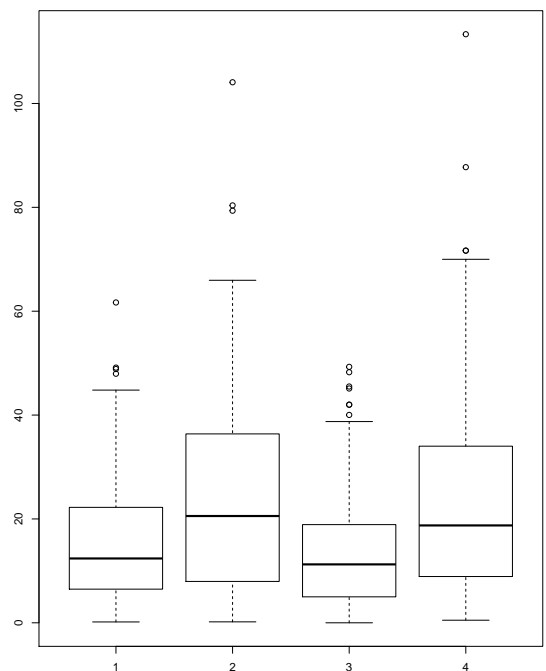

**Figure 3.** Distribution of absolute error of: MIGHTv2 average, regular average, MIGHTv2 median and regular median calculated for the Potassium (K) content in foods from the food group "Fruits".

Table 7 shows the percentage of cases when MIGHTv1/MIGHTv2 gives a smaller absolute error than the common approach using either the average or the median from all of the countries. In this

table, MIGHT with averages is compared to regular averages (measure: average), while MIGHT with medians is compared to regular medians (measure: median). From these results, it is clear that no matter the MIGHT version, if the median is used for the eligible countries, the percentage of cases when the proposed methodology gives a smaller absolute error is higher than the percentage when the average value from the eligible countries is used. This is expected because it is known that outliers affect averaging.

**Table 7.** Percentage of cases when MIGHT gives smaller error than regular averages or medians.

| Use Case/Measure | MIGHTv1 | | MIGHTv2 | |
|---|---|---|---|---|
| | Average | Median | Average | Median |
| Potassium (K) in Fruits | 84.0 % | 84.7 % | 66.9% | 76.0% |
| Sodium (Na) in Fruits | 79.9% | 88.9% | 69.4% | 82.6% |
| Sodium (Na) in Vegetables | 82.0% | 88.0% | 74.0% | 78.0% |
| Protein (PROT) in Meats | 87.0% | 88.0% | 80.9% | 82.6% |

Because the comparisons presented in Table 7 are made by reporting percentages, we further investigate if there is a statistical difference between the absolute errors obtained by MIGHTv1, MIGHTv2 and regular approach using average/median. To perform this, we compare the absolute errors obtained from the three approaches on the dataset for Potassium (K) in Fruits separately for three different countries (Italy, UK, and USA). For all comparisons, the assumptions for using the Friedman test are satisfied and the significance level is set at 0.05 [54]. The obtained $p$-values are reported in Table 8, where the comparisons are made between: MIGHTv1 with averages, MIGHTv2 with averages, and regular averages (measure: average); and MIGHTv1 with medians, MIGHTv2 with medians, and regular medians (measure: median). In the table, we can see that the null hypothesis is not rejected only in the case when we compare MIGHTv1, MIGHTv2 and regular averages for USA; in all other cases, the null hypothesis is rejected. Further, to see the specific differences among the approaches using absolute error, we perform the Nemenyi test as a post-hoc procedure on the data from the UK. The results are presented in Tables 9 and 10, where we can see that the null hypothesis is rejected only in the cases when MIGHTv1/v2 is compared to regular averages/medians.

**Table 8.** Statistical comparison between absolute error obtained from MIGHTv1, MIGHTv2, and regular averages/medians for three countries by using the Friedman test.

| Country/Measure | Average | Median |
|---|---|---|
| Italy | $5.82 \times 10^{-5}$ | $1.14 \times 10^{-4}$ |
| UK | $1.36 \times 10^{-4}$ | $5.02 \times 10^{-4}$ |
| USA | $9.90 \times 10^{-2}$ | $9.90 \times 10^{-2}$ |

**Table 9.** Results from Nemenyi post-hoc test for UK obtained by using MIGHTv1 with average, MIGHTv2 with average, and regular average.

| Average | MIGHTv1 | MIGHTv2 | Regular |
|---|---|---|---|
| MIGHTv1 | / | $5.38 \times 10^{-3}$ | $2.05 \times 10^{-4}$ |
| MIGHTv2 | $5.38 \times 10^{-3}$ | / | $9.89 \times 10^{-3}$ |
| Regular | $2.05 \times 10^{-4}$ | $9.89 \times 10^{-3}$ | / |

**Table 10.** Results from Nemenyi post-hoc test for UK obtained by using MIGHTv1 with median, MIGHTv2 with median, and regular median.

| Median | MIGHTv1 | MIGHTv2 | Regular |
|---|---|---|---|
| MIGHTv1 | / | $9.82 \times 10^{-1}$ | $3.08 \times 10^{-3}$ |
| MIGHTv2 | $9.82 \times 10^{-1}$ | / | $5.61 \times 10^{-3}$ |
| Regular | $3.08 \times 10^{-3}$ | $5.61 \times 10^{-3}$ | / |

## 4. Discussion

To compare our methodology with current state-of-the-art approaches for obtaining missing values in data tables, we decided to compare MIGHT with NMF. To apply the NMF algorithm we used the "NNLM" (Non-Negative Linear Models) [56] package for the R programming language. This implements fast sequential coordinate descent algorithms for non-negative linear regression and NMF. It is used for fast and versatile NMF. In this case, we used the previous example of Potassium (K) values in the food group "Fruits". The algorithm works on the whole set, i.e., table, and can obtain multiple missing data. To make a fair comparison of the algorithm, we apply it when there is only one missing data value. The data are looped and in each step of the loop one value is set as "NA", which implies a missing value, and then the NMF algorithm is applied on the whole set. The error is calculated by comparing the obtained value from the algorithm and the actual value. After the application of the algorithm, the missing value is calculated. At the end of the loop, all the data have gone through that process and what is left is to compare the calculated values with the actual values and to calculate the absolute error values for each one. When this has been done, the results from this method are compared with the results obtained from our methodology.

The results of the comparison are presented in Table 11 when All vs. all post-hoc type procedure (MIGHTv1) is used, and in Table 12 when One vs. all post-hoc type procedure (MIGHTv2) is used. The comparison shows that our methodology gives smaller error in around 60% of the cases on average and wins over the current state-of-the-art methodology for any type of missing values in datasets. These results are very promising. The NMF has an advantage that it can be applied to a dataset that has many missing values. Of course, as the number of missing data increases, the precision of the calculated values will decrease.

**Table 11.** Percentage of cases when MIGHTv1 gives smaller error than non-negative matrix factorization.

| Use Case/Measure | Average | Median |
|---|---|---|
| Potassium (K) in Fruits | 58.4% | 57.9% |
| Sodium (Na) in Fruits | 55.8% | 52.8% |
| Sodium (Na) in Vegetables | 52.7% | 58.0% |
| Protein (PROT) in Meats | 50.9% | 52.8% |

**Table 12.** Percentage of cases when MIGHTv2 gives smaller error than non-negative matrix factorization.

| Use Case/Measure | Average | Median |
|---|---|---|
| Potassium (K) in Fruits | 58.6% | 55.8% |
| Sodium (Na) in Fruits | 59.3% | 66.3% |
| Sodium (Na) in Vegetables | 66.0% | 69.3% |
| Protein (PROT) in Meats | 59.6% | 62.1% |

Regarding the time complexity of MIGHT, in Table 13, the execution times for obtaining the rules for all the use cases are presented. This time was calculated using a computer with a 4-core *i5* central processing unit that runs with a clock rate of 2.6 GHz. The datasets from each use case are with the following dimensions:

- Potassium (K) in Fruits — $16 \times 9$;
- Sodium (Na) in Fruits —$16 \times 9$;
- Sodium (Na) in Vegetables —$15 \times 9$; and
- Protein (PROT) in Meat —$13 \times 9$.

**Table 13.** Execution times for obtaining rules for all use cases.

| Use Case | Execution Time MIGHTv1 | Execution Time MIGHTv2 |
|---|---|---|
| Potassium (K) in Fruits | 4.5 s | 4.3 s |
| Sodium (Na) in Fruits | 3.9 s | 3.9 s |
| Sodium (Na) in Vegetables | 3.9 s | 3.6 s |
| Protein (PROT) in Meat | 2.5 s | 3.0 s |

In addition, we explored cases where MIGHT gives the greatest absolute error. Such case was obtained for the use case of Sodium (Na) in Vegetables for the food Spinach. When investigating why this happens, we found that the root of the problem was the big deviation in the data, which is presented in Table 14. A big error is bound to happened in such case because of this inconsistency. The results from MIGHT can be improved if data cleaning approaches are incorporated into the proposed methodology to produce a methodology that does not require human involvement.

**Table 14.** Big variations in the data for nutrient values of Sodium (Na) in foods from the food group "Vegetables".

| Food/Country | IT | UK | CH | BE | DK | SI | NL | USA | CA |
|---|---|---|---|---|---|---|---|---|---|
| Beet root | 10 | 66 | 58 | 58 | 43 | 58 | 70 | 78 | 77 |
| ⋮ | ⋮ | ⋮ | ⋮ | ⋮ | ⋮ | ⋮ | ⋮ | ⋮ | ⋮ |
| Spinach | 100 | 140 | 65 | 105 | 41 | 60 | 13 | 79 | 24 |

## 5. Conclusions

Focusing on the topic of quality improvement of food composition databases, we present a methodology which can improve the quality of existing FCDBs. We decided to tackle one of the most common and quickly growing problems in FCDBs, which is missing food composition data (FCD). Our proposed methodology, MIGHT, deals with the incomplete coverage of foods or nutrients leading to missing data by introducing rules for borrowing data for imputation of missing values from other FCDBs, generated with modeling based on null hypothesis testing.

We explore the idea that proper statistical analysis can decrease the error of borrowing data for imputation of missing values in FCDBs. The null hypothesis for testing is set to be "There is no effect difference between the separate datasets", the datasets being the separate national FCDBs that can possibly borrow data from each other. On the collected data, adequate statistical analysis are performed for testing the null hypothesis. After that, based on the post-hoc procedure used, our methodology goes in two directions: MIGHTv1 uses All vs. all post-hoc procedure and MIGHTv2 uses One vs. all post-hoc procedure. After the post-hoc procedure application, the rules for borrowing are generated. These rules are generated as many times as there are instances in the set used for the experiments. The final rules are a result of summarizing the rules generated by all models. When the final rules are generated, we introduce measures for calculating the value by using the rules. Our measures of choice are average and median, because of the fact that current techniques for borrowing data for imputation of missing FCD most commonly involve these two measures for calculating the nutrient value that has to be borrowed. The end result for a missing value of a specific nutrient in a food from a given country is a value obtained from a set of countries whose FCDBs are eligible for borrowing.

For future work, we plan to investigate the properties of the proposed methodology by performing sensitivity analysis using simulated data with known characteristics. We also plan to extend the

methodology to work for data imputation of more than one missing value using multivariate statistical analysis and some modification of the NMF method. On the topic of missing FCD, one of our next steps is also considering other solutions for missing data—calculating missing data from recipes [16] or estimating values from a similar food or the same food in different form. In terms of improving MIGHT, we are planning to work on an extension which will handle situations when more than one nutrient value is missing, following the idea of non-negative matrix factorization in a combination with multi-view learning [57,58].

**Author Contributions:** G.I. and T.E. proposed and designed the methodology for missing value imputation using modeling based on *null hypothesis* testing. G.I. and T.E. designed and preformed the experiments, and analyzed the data. P.K. and B.K.S. helped directly in the study. B.K.S. contributed data and materials. G.I. and T.E. wrote the paper. B.K.S contributed to the manuscript.

**Funding:** The research reported in this manuscript was supported by the project ISO-FOOD, which received funding from the European Union's Seventh Framework Programme for research, technological development and demonstration under grant agreement No. 621329 (2014–2019). This work has also received funding from the European Union's Horizon 2020 research and innovation programme under grant agreement No 769661, and the Slovenian Research Agency (research core funding No. P2-0098).

**Acknowledgments:** The authors acknowledge the financial support from both funding parties.

**Conflicts of Interest:** The authors declare no conflict of interest.

## Abbreviations

The following abbreviations are used in this manuscript:

| | |
|---|---|
| A | Analytical |
| AG | Analytical, generic |
| BE | Belgium |
| CA | Canada |
| CH | Switzerland |
| DK | Denmark |
| EuroFIR | European Food Information Resource Network |
| FAO INFOODS | International Network of Food Data Systems |
| FCD | Food Composition Data |
| FCDB | Food Composition Database |
| IT | Italy |
| K | Potassium |
| MIGHT | Missing Nutrient Value Imputation UsinG Null Hypothesis Testing |
| Na | Sodium |
| NL | Netherlands |
| NMF | Non-Negative Matrix Factorization |
| NNLM | Non-Negative Linear Models |
| POS | Part of Speech |
| PROT | Protein |
| scmamp | Statistical Comparison of Multiple Algorithms in Multiple Problems |
| SE | Sweden |
| SI | Slovenia |
| UK | United Kingdom |
| USA | United States of America |

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
