# Peer review of "MIGHT: Statistical Methodology for Missing-Data Imputation in Food Composition Databases"

_applsci, doi:10.3390/app9194111_

Round 1
Reviewer 1 Report
The missing data problem considered in the paper is very interesting. However, the paper need be rewritten to focus on the proposed method and reduce details which are not directly related to the method. My comments are as follows.
The post-hoc procedures, MIGHTAv1 and MIGHTv2 need to be explained clearly, say what is “All vs. all” and “One vs. all” (see page 8) as this is the very important step in the proposed method. Authors should give some comments on how to select the alpha value. For a given alpha according to the MIGHT method it is possible that for some countries there is no country that are eligible for borrowing, say there is no “0” in a particular row for Table 5. Then, select a smaller alpha or else? The matrix in Table 5 should be symmetric? There are duplicated information and too much details on well know results. For example Table 2 and Table 3 are essentially the same thing. Too much details about “what is a hypothesis test”.
Author Response
The missing data problem considered in the paper is very interesting. However, the paper need be rewritten to focus on the proposed method and reduce details which are not directly related to the method. My comments are as follows.
The post-hoc procedures, MIGHTAv1 and MIGHTv2 need to be explained clearly, say what is “All vs. all” and “One vs. all” (see page 8) as this is the very important step in the proposed method.
-Since we only had explanations on which post-hoc procedures are used in MIGHTv1 and MIGHTv2 (Lines 403-410), we included additional explanation for MIGHTv1 and MIGHTv2 with references (Lines 302-312)
Authors should give some comments on how to select the alpha value.
-Explanation for selecting the alpha value are added. (Lines 270-274, 395-396)
For a given alpha according to the MIGHT method it is possible that for some countries there is no country that are eligible for borrowing, say there is no “0” in a particular row for Table 5. Then, select a smaller alpha or else?
-Our significance level α is set at 0.05 which is the most commonly used value for a significance level [1]. While testing our methodology in the four data sets we had, and all 8 experiments that we conducted we did not experience such case, i.e., the null-hypothesis is rejected in all cases - no country is eligible for borrowing. However, if that were to happen, if all p – values are lower than α, it would lead us to select a smaller α value
In [2], Montgomery states that having a large sample size results in small b-risk and small sample size results in large b-risk, i.e. large risk to fail to reject the null hypothesis, which in our case means larger risk of having all “0” in our rules. Since we are working with small data sets – small sample sizes the scenario of always rejecting the null-hypothesis is highly unlikely, and changing up the level of significance α is something to consider if working with significantly larger sample sizes – data sets.
In [3] the authors go into details on the relation between the level of significance and the sample size.
“The general procedure in hypothesis testing is to specify a value of the probability of type I error and then to design a test procedure so that a small value of the probability of type II error is obtained. Thus, we can directly control or choose the risk. The risk is generally a function of sample size and is controlled indirectly. The larger is the sample size(s) used in the test, the smaller is the risk.”[2]
The matrix in Table 5 should be symmetric?
-Table 5 (now Table 4) is not bound to be symmetric, because when using One vs. All post-hoc test the rules are not symmetrical. We have changed the table so it represents both versions of the methodology.
There are duplicated information and too much details on well know results. For example Table 2 and Table 3 are essentially the same thing. Too much details about “what is a hypothesis test”.
-We tried to delete any duplicated information in the manuscript. We agree that Table 2 and 3 represent the same thing, thus Table 2 has been deleted. The information about “what is hypothesis test” has been reduced.
[1] Cowles, M. and Davis, C., 1982. On the origins of the. 05 level of statistical significance. American Psychologist, 37(5), p.553.
[2] Montgomery, D.C., 2009. Statistical quality control (Vol. 7). New York: Wiley.
[3] Montgomery, D.C. and Runger, G.C., 2007. Applied statistics and probability for engineers, (With CD). John Wiley & Sons.

Reviewer 2 Report
Comments and suggestions are in the attached document.

Author Response
Title: the title is not aligned with the content of the document. The expectation is a computer intensive application of some advanced analytics for borrowing data. However, reading the paper, there is no trace about this. The implementation is on R and it seems also off-line, so it is a traditional analysis. Personally, I will revise the title to make coherent the expectation and what written in the documents.
- The title has been revised:
“MIGHT: Statistical Methodology for Missing-Data Imputation in Food Composition Databases” Abstract: overall the abstract is well sounded with the content
-The abstract did not have any changes.
The introduction is built upon basically one reference cited more time (apart from the list in row
41). However, some statements need to be supported by extant literature, e.g. 32-33 about the
limitations, 42-47 about the second limitation (missing data). In particular this last one need to be supported by literature since it is the one on which the authors are focusing on.
-Added references about the FCDBs limitations, and also included other important references in throughout the manuscript.
The introduction also misses how the paper is structured, so the final part must be more focused on this and rows 88-91 in section 2 must be moved in introduction
-Structure of paper is added at the end of the Introduction section.
Figure 1 is not explicative wither useful to the paper. Data models are very complex and are codified in different languages, but they could also be represented graphically. Since this paper is not focused on this topic, I suggest two options: provide the graphical representation of a part of the data model by EuroFIR or remove the figure since it is not propaedeutic to the paper.
-We agree that Figure 1 was not crucial in understanding the paper so it has been removed from the manuscript.
Subsection 2.3 is the core of the paper since it presents the so-called methodology (I have some
thoughts about this being a subsection as well, maybe an entire section is preferred), moreover:
The current presentation of the methodology is not linear and is difficult to read, I suggest reviewing the writing style In my opinion, there is a need for a figure explaining the whole methodology. This figure could represent the methodology from an upper level, then different subsections could be used to enter in the details of each step The statistical checking of different assumptions is a bit confusing since the authors state that it could be used this test, this other one, or another one and so on. However, the authors do not provide any indication on how to select one or another one.
-We agree that subsection 2.3. is the core of the paper, but since the format of the Journal of Applied Sciences has Materials and Methods in one section, the methodology cannot be separated as a new section.
The writing style has been reviewed and revised. Figure 1 is added in order to explain the methodology. The checking of the three conditions for choosing the appropriate test is changed in order to be more comprehensible. In section 2.3. (Lines 250-262) we are stating all the possible statistical test that can be used for checking these conditions. We reference [1], where the authors explain in details why checking these conditions is important and go into details on how these conditions are checked. Section 3 (Results) is too long. The section results should be limited to the numerical results
obtained by the application of the methodology; instead, section 4 (Discussion) must provide
critical insights on the results (in fact, right now, section 4 discussion is essentially empty with no
additional messages)
-Section 3 (Results) is shortened, parts of the section are transferred to section 4 (Discussion).
Section 4 (Discussion), see point 5, is too short. In this section there is the real novelty of what
proposed and so this must be extended including all the reasonings stated in section 3 results.
-Section 4 (Discussion) is extended with parts of section 3 (Results) and additional explanation on the importance of this work.
As minor issues:
row 441, word “Form” seems to be wrong, maybe “From”
- The sentence has been changed
BS EN 16104:2012 is cited in the text, but there is no reference
-reference for BS EN 16104:2012 is added
Overall, I’m also concerned also about the missing, within the citation/reference, of the works by Montgomery, statistician and expert in design and analysis of experiments.
-References from the work by Montgomery have been added – references 41 and 42 in the manuscript (Lines 272 and 273)
[1] Eftimov, T., Korošec, P., Potočnik, D., Ogrinc, N., Heath, D. and Seljak, B.K., 2017. How to perform properly statistical analysis on food data? An e-learning tool: Advanced Statistics in Natural Sciences and Technologies. Book chapter-Science within Food: Up-to-date Advances on Research and Educational Ideas.

Round 2
Reviewer 1 Report
The proposed method only takes three or four pages, but the paper is more than 18 pages. If unnecessary information is removed the paper will look better.
Author Response
We tried to remove unnecessary information as much as possible. Some rearrangements were made in the manuscript in order for it to be more understandable to the reader. The section with the methodology (2.3.) was improved and extended, also additional information was added to Figure 1 in order to guide the reader through understanding the methodology.
Currently out of all 17 pages of text, the MIGHT - Missing Nutrient Value Imputation UsinG Null Hypothesis Testing (which contains the explanation of the methodology) and the Results sections take 9 and a half pages, the other sections are brought to a minimum.
Reviewer 2 Report
Overall the manuscript has been improved and the provided comments have been added. I appreciate the effort made by the authors.
Currently, I feel the study has the right soundness and potentiality to be published in Applied Sciences.
I would like to stress Fig.1 that has been added. I think this provides real additional value to the paper. However, in this current form, it does not emphasise the effort the authors made to realise MIGHT. I think that some more details in Fig.1 (or reference to the different parts/sections of the paper) could be useful. Fig.1 could be of real help to the reader to move within the content.
I would like only to underline that there could be two minor errors:
line 10 (abstract): the reference to the standard must be transformed into a simple "BS EN 16104 published in 2012". In the present form, it seems to be a citation that is better not to include in the abstract (also at line 100 is better two remove the round brackets since it is not a citation) line 32 (introduction): it seems there is a residual of a copy&paste action since there is a capital letter "Two limitations" in the middle of the sentence.
I think that once the minor issues will be checked, the paper has a sufficient level for the publication on Applied Sciences.
Author Response
I would like to stress Fig.1 that has been added. I think this provides real additional value to the paper. However, in this current form, it does not emphasise the effort the authors made to realise MIGHT. I think that some more details in Fig.1 (or reference to the different parts/sections of the paper) could be useful. Fig.1 could be of real help to the reader to move within the content.
Figure 1 has been modified, more details have been added to it and references to different parts of the paper.
line 10 (abstract): the reference to the standard must be transformed into a simple "BS EN 16104 published in 2012". In the present form, it seems to be a citation that is better not to include in the abstract (also at line 100 is better two remove the round brackets since it is not a citation) line 32 (introduction): it seems there is a residual of a copy&paste action since there is a capital letter "Two limitations" in the middle of the sentence.
Thank you for pointing it out, all the changes were made.